

# 1  Past and future climate analysis at regional scale: the case study of the Campania

# 2  Region, Italy

Giugliano Giuseppe[1], Villani Veronica[1], Barbato Giuliana[1], Schiano Pasquale[1], D'Ambrosio Antonio[2],
Cau Piero[2], Onorati Giuseppe[2], Mercogliano Paola[1]
[1]Regional Models and geo-Hydrological Impacts (REMHI) Division, Fondazione Centro Euro-Mediterraneo sui Cambiamenti
Climatici, Caserta, 8100, Italia
[2]ARPA CAMPANIA, UOC Reti di monitoraggio e Centro meteorologico e climatologico, Napoli, Italia
*Correspondence to*: Villani V. (veronica.villani@cmcc.it)
**Abstract**. The present paper reports a detailed analysis of the observed and expected climate conditions
over the Campania Region, located in the South of Italy. Campania, as part of the Mediterranean area, is
already testing relevant impacts related to climate change; evaluation is expected to support local
stakeholders in the risk assessment for different relevant sectors but also support the adaptation process.
Due to the above mentioned goal the analysis of climate condition is carried out through the evaluation
of ETCCDI climate indicators largely used to characterize the frequency and intensity of extreme climate.
Analysis is performed both starting from the observed data, provided by various in situ meteo-climatic
monitoring networks of the Campania Region managed by local Civil Protection, and using a high
resolution regional climate models. Particularly, a systematization of historical data was first carried out
by performing the data completeness test and later the climate quality check and homogenization test of
the complete time series, which ensure a correct evaluation of climate indicators over the region.
Furthermore, for the calculated climate indicators, the variations expected on the territory for 4 future
periods and under 2 different IPCC climate scenarios were evaluated, starting both from the data of the
COSMO CLM regional model, and from the ensemble mean of the EURO-CORDEX models.



## 1 Introduction

As part of the POR FESR Campania 2014/2020 Project "Services and Gov Advanced Climate Products" an agreement was signed between the Regional Agency for Environmental Protection of Campania and CMCC srl, CMCC Foundation's spin off, to produce information and climate analysis to support the planning activities of the Campania Region. This is in order to provide citizens, businesses and other bodies with a detailed knowledge of the climate changes observed and expected in the region in the coming years.

In particular, this information is very useful in the planning of interventions on the territory by, for example, the Municipalities themselves to support public and private users to define adequate adaptation and risk analysis actions for different sectors of interest (such as for example, tourism and agriculture, health) from the regional to the municipal scale.

Therefore, the present work aims to describe the climatic characteristics of the Campania Region, both as regards the observed local climate and as regards the expected future scenarios. This description is based on the use of observational datasets and high-resolution climate projections currently available on the Italian territory. The climate analysis was carried out using indicators deemed relevant to assess the main local impacts of climate change (Karl et al. 1999, Peterson et al. 2001). Particularly, as regards the analysis on the observed data, it was first conducted a systematisation of the station data by performing the data completeness test, and later the climate quality check and homogenization test of the complete time series (ISPRA 2012, 2013; Fioravanti et al., 2016). This allowed the calculation of climatic indicators on the observed data of the Campania Region. Furthermore, for the same climate indicators, the expected variations in the territory were assessed starting both from the data of the COSMO CLM regional climate model of the CMCC Foundation at the resolution of 8 km available on Italy, and of the ensemble mean of the EURO-CORDEX models at maximum resolution available, about 12 km, considering the two different IPCC scenarios RCP4.5 and RCP8.5 (IPCC, 2014a; Moss et al, 2010). The study of the climate implies, by definition, the use of long time scales; in particular, the WMO (WMO, 2007) establishes in 30 years the standard length on which it is necessary to perform statistical analyses that can be considered representative of the average and extreme climate regimes of a selected geographical area. For this reason, variations of future climate (in terms of average and extreme patterns), due to different climate-changing gas concentration scenarios, have been compared to the reference climate by comparing two periods, each of 30 years length.

The paper is organised as follows: the *Methodology* section describes the methodology developed in this work. In particular, completeness test and data quality check are described in general form in subsections 2.1 and in detail in subsections 2.1.1, 2.1.2 and 2.1.3. Subsections 2.2 provides a general overview of the homogenization process and, subsequently, the four homogenization methods used are described in detail in subsections 2.2.1 to 2.2.4. In subsection 2.3 the climate analysis carried out in this work on the Campania region is described. The main results obtained are commented in Sections 3 and 4. Finally, Section 5 briefly summarize the results obtained and provides the conclusions of this work.



## 2 Methodology

The present work aims to describe the climate characteristics of the Campania Region starting from the observed data provided by the various meteo-climatic monitoring networks of the region, and to analyse the region's climate change scenarios using high-resolution regional climate models currently available on the Italian territory. Before being able to perform the climate analysis it was necessary to carry out a systematisation of the station data, which consists of execution of data completeness test, climate quality check and homogenization test of the complete time series (ISPRA 2012, 2013; Fioravanti et al., 2016).

### 2.1 Completeness test and data quality check

The completeness check of the station data series consists in verifying that there is a minimum necessary number of data on which to perform a climate analysis. In particular, it consists in checking for the presence of at least 75% of available data, as the presence of missing data can lead to insignificant, strongly distorted and/or even incorrect analysis (ISPRA 2012, 2013). Instead, quality checks are defined as those techniques and activities that are used to satisfy the required quality requirements. The main purpose is the detection of missing data, reporting and possibly correcting errors in order to ensure the highest standard of accuracy for the optimal use of data by users. The most important prerequisites for performing a rigorous climate analysis include the following:

1. checking the completeness of the series;
2. basic integrity checking and outlier management;
3. the check on the homogeneity of the series, useful to ensure that the variations present are due exclusively to climatic factors (Conrad et al. 1950). This last check essentially concerns the identification of any breakpoints that represent the instant in time in which the series begins to manifest a perturbation (which involves a variation on the average value of the series).

To correctly evaluate all these issues, various methodologies have been implemented, starting from those developed by ISPRA (Fioravanti et al., 2016), mostly of a statistical nature.

### 2.1.1 Completeness test

The study of the climate implies, by definition, the use of long time scales; in particular, the World Meteorological Organization (WMO 2007) establishes in 30 years the standard length on which to carry out statistical analysis that can be considered representative of the climate of a certain area. Therefore, before performing climate analysis using the station data series, they must be subjected to a completeness check, which consists in verifying the availability of at least 75% of data over 30 years, as the presence of missing data can lead to insignificant, strongly distorted and/or even incorrect analysis (ISPRA 2012, 2013). In the present case, as reported in the introduction, the time series are available over a period of 20 years, from 2001 to 2020, thus shorter than the standard 30-year period. Nevertheless, The Role of Climatological Normals in a Changing Climate (WMO, 2007) reported that, for most mean and sum parameters, 10–12 years of data provided a predictive skill similar



to that from a standard 30-year period, and "while such short periods cannot be considered to be climatological standard
normals or reference normals, they are still useful to many users, and in many cases, there will be benefits to calculating such
averages operationally." In our case, with longer than 10-12 years' time series, we chose to carry out the completeness test on
103 precipitation stations and 45 average temperature stations, considering complete the stations with at least 60% of data
available over 30 years, thus requiring the presence of at least 18 complete years, where complete means that every year it has
at least 75% of not null data inside it.

**2.1.2 Basic integrity test**
The basic integrity tests are used to search within a series for the presence of repeated, suspicious or impossible values;
moreover, these tests identify equal years to each other and equal months both in different years and in the same years. In this
section, the tests for precipitation and temperature will be described in Table 1.

| Check | Condition to invalidate | Analyzed variables | Comments |
|---|---|---|---|
| Repeated values | 10 or more identical consecutive values. | Tmax, Tmin, Prec | Missing data is not considered for temperatures. Precipitation missing data and zeros are not considered. |
| Zeros persistence | 180 or more consecutive null values. | Prec | |
| Duplicate years | All daily values of one year equal to all values of another year. | Tmax, Tmin, Prec | For the precipitation the presence of at least 5 not null values is required. |
| Duplicate months | Within the same year: all the daily values of a month equal to all the values of another month. In different years: all daily values of a month equal to all values of the corresponding month. | Tmax, Tmin, Prec | For the precipitation the presence of at least 5 not null values is required. |
| Tmax equal to Tmin | Tmax equal to Tmin for 10 or more consecutive days. | Tmax, Tmin | |
| Null values of Tmax and Tmin | Tmax and Tmin both equal to 0 °C. | Tmax and Tmin | Identifies incorrectly used zeros in place of missing data. |
| Impossible values | Tmax>50°C; Tmax<-30°C Tmin > 40°C; Tmin< -40°C Prec>800mm; Prec <0 mm | Tmax, Tmin, Prec | |


99                                       **Table 1: Basic integrity test.**






### 2.1.3 Identification test of anomalous values


Two different approaches are used to identify anomalous values in a series: the first is concerned with finding any jumps in
the series and the second is of a climatological type. To find jumps in the series, tests defined as "interval check" are used (Gap
check, Table 2). Climatological controls, on the other hand, are different in terms of temperature or precipitation, for the first
the z-score is used (Table 2) which is the most common method for identifying anomalous values in the meteo-climatic data,
it indicates how many standard deviations a certain value is is positioned with respect to the average, the maximum value it
can take is indicated by the following equation (Shiffler, 1988):
$z_n = \frac{n-1}{\sqrt{n}}$ ,    (1)
where n is the number of samples; this test is used because it can be assumed that the temperature has a roughly normal
distribution. This methodology can only be applied to sufficiently long series, because in the case of even slightly asymmetric
distributions it could cause an excessive number of false positives. The test first normalised the data using moving mean and
variance, relating to the period under examination, these two values are calculated on all the data that fall within a 15-days
window centred on the day we want to examine. In the case of precipitation, a percentile-based test is applied. For each day of
the year a 29-days window is defined centred on the day in question and the $95^{th}$ percentile of the distribution of all values
other than zero that fall within the window is calculated, over the entire available period. Over the entire period in question, in
the 29-days moving window, the presence of at least 20 not null values is required. The choice aimed at using percentiles for
precipitation is due to the fact that this variable does not have a normal or Gaussian distribution, therefore, the standard
deviation would be extremely influenced by the values found in the tail of the distribution.





| Check | Condition to invalidate | Analyzed variables | Comments |
|---|---|---|---|
| Temperature Gap check | Difference ≥ 10 °C between two consecutive values in the monthly temperature distribution. | Tmax, Tmin | |
| Relative Humidity Gap check | Variations ≥ 30% between one data and another are to be considered incorrect (Nyckowiak et al., 2010). | RH | |
| Precipitation Gap check | Difference ≥ 300 mm between two not null values in the monthly distribution of Prec. | Prec | |
| z-score | $\|z\| \geqslant 6$ (Fioravanti et al., 2016) | Tmax. Tmin | At least 100 values are required over the entire period in the 15-days window. |
| z-score | $\|z\| \geqslant 5$ (Lanzante, 1996) | RH | |
| 95th percentile (T medium ≥ 0 or not available) | Prec 9 times the 95th percentile | Prec | At least 20 not null values are required over the entire period, in the 29-days window. |
| 95th percentile (T medium < 0) | Prec 5 times the 95th percentile | Prec | At least 20 not null values are required over the entire period, in the 29-days window. |

**Table 2: Identification test of anomalous values.**

**2.2 Homogenization**

After quality check, the time series must undergo the homogenization process (Vezzoli et al. 2012). A series is homogeneous if its variability depends exclusively on climate factors, however there may also be causes of another nature that lead to a lack of homogeneity of some kind (ISPRA 2012), such as the change of the instrumentation, changes in the position of the station or change of the environment surrounding the station (Alexandersson 1986). The moment in which time series show a perturbation is called a breakpoint, and the identification and removal of these points is not a secondary aspect. The homogenization of hourly and daily data is very complex, due to the high variability of the data and the presence of extreme values that pose many problems since they are, by definition, rare events. A further complication derives from the fact that different methods can produce discordant results for the same time series of data; for this reason it is widely believed that a careful study of inhomogeneities of a series should take into consideration the result of different statistical methods (Wijngaard



et al. 2003). Most of the statistical methodologies used are based on the comparison between the series under examination and
a reference time series. Since in reality it is almost impossible to find perfect series, it is a common practice to resort to artificial
reference series constructed by appropriately combining the climate signal of time series of neighboring stations sufficiently
correlated with the candidate (Reeves et al. 2007). In this work will be analyzed and developed four homogeneity tests which
are: the Standard Normal Homogeneity test, the Buishand Range test, the Pettitt test and the Von Neumann Ratio test.
Based on what has been said, a time series can be classified:
-     Potentially homogeneous: if the time series is homogeneous for at least three out of four tests;
-     Unsure: if the time series is homogeneous for two out of four tests;
-     Suspect: if the time series is homogeneous for one or no in four tests.

### 2.2.1 Standard Normal Homogeneity test

The Standard Normal Homogeneity test (SNHT) was applied to climate data by Alexandersson in 1986. This method provides
valuable noise reduction in the series and illustrates the main idea of testing relative homogeneity.
It proceeds by defining a new set of standardised ratio calculate following Eq. (2):
$$z_i = \frac{q_i - \bar{q}}{s_q},\qquad\qquad (2)$$

where $\underline{q}$ is the arithmetic mean of ratio $q_i$, $s_q$ is the series standard deviation. The new series $z_i$ has exactly mean 0 and standard
deviation 1; this is the biggest assumption of the test, because it is possible only for certainly homogeneous data as it is also a
single breakpoint the standard deviation could suffer a weak bias (Reeves et al. 2007). Considering the initial assumption valid,
the null and alternative hypothesis can be defined; the null hypothesis $H_0$ is defined following Eq. (3):
$$H_0 : Z \in N(0,1), \forall i ,\qquad\qquad (3)$$

and the alternative hypothesis $H_1$ is defined following the Eq. (4):
$$H_1 : \begin{cases} For\ some\ 1 \leq v < n\ and\ \mu_1 \neq \mu_2\ we\ have \\ \qquad Z \in N(\mu_1, 1),\ for\ i \leq v \\ \qquad Z \in N(\mu_2, 1),\ for\ i > v \end{cases} ,\qquad\qquad (4)$$

In the null hypothesis, it is assumed that the Z has a normal distribution with 0 mean and deviation standard 1. With this it is
assumed that the sequence of ratios is described by normal distribution and that the possible breakpoint is only one and consists
only of a displacement from the mean value. In the alternative hypothesis, instead, $\mu_1$ and $\mu_2$ are the mean values of normal
distributions which all have standard deviations equal to unity. The standard relationship likelihood technique (Lindgren 1968)
calculated following the Eq. (5), can be used to answer this question.
$$Max_{\mu_1 \mu_2 v} = \frac{(2\pi)^{\frac{-n}{2}} e^{\frac{-1}{2}(\sum_{i=1}^{v}(z_i - \mu_1)^2 + \sum_{i=v+1}^{n}(z_i - \mu_2)^2)}}{(2\pi)^{\frac{-n}{2}} e^{\frac{-1}{2}\sum_{i=1}^{n} z^2_i} } > C ,\qquad\qquad (5)$$



Obviously the critical values of the test will depend on the number of samples that are used but are chosen to reference the
ninetieth and ninetieth-fifth percentile of the distribution, therefore, these two values are taken as threshold since they are in
the right position of the distribution in order not to miss false positive and detect false negative values. The reconstruction of
the time series occurs by estimating each element of the series as a weighted average of a prescribed number of closest available
data. The weights to be applied are calculated through the inverse distance between the observation sites (Guijarro 2018).
In the first instance it was common opinion to think that the SNHT identified discontinuities in series even shorter than five
years, subsequent studies have shown the need for have at least available time series with 20 years of processed data,
furthermore it has been noted the low propensity of this test to identify breakpoints in the middle years of the series, it performs
better at extreme (Alexandersson et al. 1996, DeGaetano et al. 2006 ).

### 2.2.2 Buishand Range Test

Suppose you want to test the homogeneity of a time series, under the null hypothesis it is assumed that the elements of time
series have the same mean. Generally the form of alternative hypothesis is vague, since there isn't prior information about the
change in the mean values; usually, some assumption are made about the distribution of the data, most test require them to be
independent, but this is not a serious limitation, as tests are usually performed on consecutive seasonal or annual values. Being
daily data, the test developed in this work derives from the case in which the element of time series are stochastically
independent and have a normal distribution, even if the test can still be applied when there are slight deviation from the normal
distribution (Buishand 1982, Militino et al. 2020).
Assuming a change in the mean at time $m$, we can write Eq. (6)
$$E(Y_i) = \begin{cases} \mu, i = 1, \dots, m \\ \mu + \Delta, \ i = m + 1, \dots, n \end{cases},$$   (6)
This model assuming an average shift of magnitude $\Delta$ after $m$ observations; this homogenization test based on the cumulative
deviation from the mean, computed by Eq. (7)
$$S_k^* = \sum_{i=1}^{k}(Y_i - \overline{Y_\iota}), k = 1, \dots, n \ ,$$   (7)
For a homogeneous record the result of Eq. (7) would be expected to oscillated around zero since there is no systematic error
in the deviations of $Y_i$ from the mean value $\overline{Y_\iota}$. On the other hand, if $\Delta < 0$ in the Eq. (6) many of the $S_k^*$ are positive because
$Y_i$ tend to become larger than the mean if $i \leq m$, and smaller if $i > m$. To obtain a scaled version of the partial sums divide
the $S_k^*$ by the standard deviation of the sample obtaining the Eq. (8)
$$S_k^{**} = \frac{S_k^*}{D_y},$$   (8)
The value of $S_k^{**}$ is not sensible to linear transformations of data and therefore the homogeneity tests are based precisely on
the latter. Another important statistic using to test homogeneity is the range described from the Eq. (9)

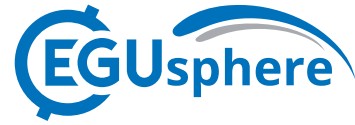

$R = max_{0 \leq k \leq n} S_k^{**} - min_{0 \leq k \leq n} S_k^{**}.$                                                                      (9)
The critical values for this test are obtained from the Kolmogorv-Smirnov statistics (Smirnov 1939) and subsequently taken
up by various author to test the null hypothesis against the alternative one; the coefficients are three and they are 0.1, 0.05,
0.01 (Gail et al. 1976), in this work we have chosen to use the value 0.05 as critical beyond which the series is not
homogeneous. Unlike the SNHT, the test currently described is able to very quickly identify the breakpoints found in the
central areas of time series (Militino et al. 2020).

**2.2.3 Pettitt test**
Consider a sequence of random variables $X_1, X_2, \ldots, X_t$ then the sequence is said to have a change-point at $\tau$ if $X_t$ for $t =$
$1, \ldots, \tau$ have a common distribution function $F_1(x)$ and $X_t$ for $t = \tau + 1, \ldots, T$ have a common distribution function $F_2(x)$
and $F_1(x) \neq F_2(x)$. We consider the problem of testing the null hypothesis of "no-change", $H: \tau = T$, against the alternative
of "change", $A: 1 \leq \tau \leq T$, using a non-parametric statistic. We make no assumption about the functional forms of $F_1$ and $F_2$
except that they are continuous (Pettitt 1979).
The Pettitt test is built following the Eq. (10)
$D_{ij} = sgn(X_i - X_j),$                                                                                                                      (10)
where $sgn(x) = 1 \, if \, x > 0, 0 \, if \, x = 0, -1 \, if \, x < 0$, then consider the Eq. (11)
$U_{t,T} = \sum_{i=1}^{t} \sum_{j=i+1}^{T} D_{ij},$                                                                                              (11)
The statistic is equivalent to a Mann-Whitney statistic for testing that the two samples come from the same population (Mann
1945, Mann et al. 1947). The non parametric statistic is defined by Eq. (12)
$K_T = max_{1 \leq t \leq T} |U_{t,T}|,$                                                                                                        (12)
And for change in one direction, the statistics became Eq.(13) and Eq.(14)
$K^+{}_T = max_{1 \leq t \leq T} |U_{t,T}|,$                                                                                                    (13)
$K^-{}_T = -max_{1 \leq t \leq T} |U_{t,T}|,$                                                                                                   (14)
It should be noted that, on the null hypothesis $H$, $E(D_{ij}) = 0$ and the distribution of $U_{t,T}$ is symmetric about zero for each $t$.
Thus $K^+{}_T$ and $K^-{}_T$ have the same null distributions.
Pettitt statistic values tend to be higher at the points that divide the time series in two parts with more pronounced differences
between them. If the test maximum exceeds a certain threshold, a breakpoint is detected. Each of the two parts of the time
series is examined separately for the identification of further breakpoints in the same way (Kaysely et al. 2005). This test uses





a remarkably stable distribution and provides a robust change point test resistant to outliers (Pettitt 1980b, Wijngaard et al.
2003). Such as the Buishand Range test, also this test is more skilled to find breakpoints in the central parts of the time series.

**2.2.4 Von Neumann Ratio test**
The Von Neumann Ratio test is defined as the ratio of the successive mean square difference, from year to years, with respect
to the variance (Von Neumann 1941); assumes under the null hypothesis that the samples of the series are independent and
identically distributed, the alternative hypothesis instead asserts that the series is not randomly distributed. This test does not
give any specification on the position of the possible breakpoint, for this reason this test is complementary to the previous ones
(Wijngaard et al. 2003)
$N = \frac{\sum_{i=1}^{n-1}(Y_i - Y_{i+1})^2}{\sum_{i=1}^{n}(Y_i - \bar{Y}_t)^2},$  (15)
If the sample contains a breakpoint that is not significant for the homogeneity of the series then the value of $N$ will be less than
2 (Buishand 1982), instead, if there is a rapid variation in the mean, the value of $N$ will tend to increase beyond the value 2
(Bingham et al. 1981).

**2.3 Climate analysis**
The climate analysis conducted in this work allows both to characterise the climate variability observed in the recent past at a
local level, identifying for example a change trend already underway for some specific characteristics of the climate, and to
evaluate, always locally, the climate changes expected in the future due to climate change on the basis of different scenarios
disclosed by the IPCC (IPCC, 2014a; Moss et al, 2010). This analysis was carried out using the indicators considered relevant
for the study of variations in intensity and frequency of extreme events, defined as events that differ, in their characteristics,
substantially from the climatological average of the area over a reference period. The indicators most used to describe the
intensity  and  frequency  of  occurrence  of  extreme  events  are  those  defined  by  the  ETCCDI
(http://etccdi.pacificclimate.org/index.shtm); they relate to various atmospheric variables, but those most commonly used in
literature concern precipitation and temperature. Therefore, these indicators make it possible to describe the variation of the
climate both in terms of average trends (variations on a seasonal and annual scale) and in terms of extremes (heat waves, very
intense rains). This last characteristic makes climate indicators a tool widely used in the literature as a proxy for the study of
variations in the characteristics (frequency and intensity) of particular impacts (EEA 2009; EEA 2018; EEA 2019; Mysiak et
al., 2018) that the climate change determines on specific sectors of interest, in order to allow, for example, the evaluation of
adaptation strategies (Karl et al., 1999; Peterson et al., 2001). The climate indicators, used in this work to describe the climate
variability of the Campania Region, are reported and defined in Table 3. The calculation of the climate indicators for the



Campania Region was carried out starting from 95 observed time series of precipitation and 38 of mean temperature, whose
data are available for the period 2001-2020. For the same climate indicators, the expected variations in the territory were
assessed starting from the data of the COSMO CLM regional climate model (Rockel et al. 2008) at the horizontal resolution
of about 8 km, forced by the global model CMCC-CM (horizontal resolution of about 80 km) (Scoccimarro et al. 2011),
adopting the configuration developed over Italy by the CMCC Foundation. This configuration has shown a good ability to
represent climate indicators on Italy both in terms of average and extreme values (Bucchignani et al. 2016, Zollo et al. 2016),
both with respect to the E-OBS observation dataset (Haylock et al. 2008) and to some available regional datasets. Furthermore,
the changes expected in the Campania region have also been assessed using the ensemble mean of the EURO-CORDEX
models at maximum resolution available, about 12 km. More information about the EURO-CORDEX initiative is available at
the following link http://www.euro-cordex.net. Furthermore, thanks to the use of an ensemble of climate models, it is also
possible to associate the expected climate changes with an uncertainty analysis, a very important element for climate adaptation
and risk analysis (Von Trentini et al., 2019). The expected climate changes were analysed in four different periods: 2021-2050,
2031-2060, 2051-2080 and 2071-2100, compared to the reference period 1981-2010, considering the two different IPCC
scenarios RCP4.5 and RCP8.5. These calculations make it possible to provide a consistent picture of the current climate and
the expected climate variations as a result of climate changes of an anthropogenic nature in the Campania Region.

|  | **Acronym** | **Definition** |
|---|---|---|
| **Mean Temperature Indicators** | **TG** – Mean Temperature (° C) | Average of the mean daily temperature. |
| **Precipitation Indicators** | **RX1DAY** - Maximum 1-day precipitation (mm/days) | Maximum 1-day precipitation. |
|  | **R20** - Days with intense precipitation (days) | Number of days with precipitation greater than 20 mm. |
|  | **RR1** - Rainy days (days) | Number of days with daily precipitation greater than or equal to 1 mm. |
|  | **CDD** - Consecutive Dry Days (days) | Maximum number of consecutive days with daily precipitation less than 1 mm. |
|  | **R95PTOT** - Fraction of precipitation on very rainy days (%) | Precipitation fraction due to precipitation greater than the 95th percentile* of the precipitation. |
|  | **PRCPTOT** - Cumulative precipitation on rainy days (mm) | Cumulated (sum) of precipitation for days with precipitation greater than or equal to 1 mm. |
| The symbol * indicates that the percentile was calculated on the reference period considered for the calculation of the threshold. | | |

**Table 3: Acronyms and definitions of the climate indicators considered in this work.**





## 3 Results: observed climate variability

This section is dedicated to the results of quality check and homogenization methods of the time series, furthermore, the results obtained from the calculation of indicators on the corrected and homogenised time series are reported. A framework of the local characteristics of the climate will therefore be provided here, both in terms of average values and in terms of extreme values. All series found to be complete (Subsection 2.1) were corrected using the quality check described in subsections 2.1.2 and 2.1.3. The results obtained from the quality check showed for the precipitation that only the Caiazzo station did not pass the persistence test of zeros. For the mean temperature, as there is no quality check, all the series were considered correct and subjected to the subsequent homogenization procedure. Indeed, following the quality check carried out on the station data series that have passed the initial completeness test, each time series was evaluated for the homogeneity (Subsection 2.2) and it was found that the precipitation time series are all potentially homogeneous except for San Mauro, Napoli Capodimonte, Pozzuoli, Castel Volturno, Colle Sannita, Luogosano, Avella, Ercolano, Castel Franco in Miscano which are resulted doubtful, but applying the SNHT it is clear that the climate signal of these stations is consistent and therefore the related data series can be considered homogeneous. For mean temperature all stations have turned out potentially homogeneous since they are homogeneous for at least three out of four tests. Through the homogenization process, the data series were reconstructed to be used for the calculation of the ETCCDI indicators listed in Table 1.

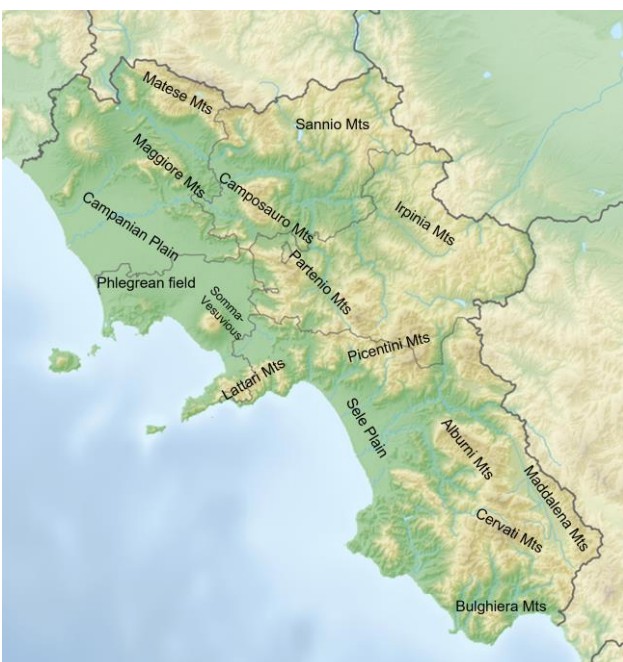

**Figure 1: Physical representation of Campania Region, Italy.**





The results obtained from the calculation of the indicators on the homogenised and reconstructed time series of precipitation
and mean temperature are exposed below. In Table 4, the average value, on annual and seasonal scale, of the climate indicators
of interest, calculated for the period 2001-2020 are shown. The different seasons are respectively the winter, called DJF; spring,
called MAM; summer, called JJA; and autumn, called SON. Furthermore, for both the annual and seasonal scales, the spatial
variability with respect to the average value of each climatic indicator considered is shown. Figures 2, 4 and 5 are shown
below, which represent the results relating to the indicators considered to be of greatest interest. Figure 2 shows the numbered
list of mean temperature stations, each number uniquely identifies the name of each station shown in the maps below. The
Figures below show the results relating to the single station, each value is averaged over the entire available period and
represented by a colored circle with the unique number of each station reported in Figure 2. The colour of the circle is relative
to the value assumed by the indicator for each station. On an annual scale (Figure 2), on the 38 stations it is noted that those
located in the innermost areas of the region and higher in altitude register a lower mean temperature, while the stations located
in the plains have a higher mean temperature. In Table 4, looking at the seasonal scale, it can be seen that the autumn season
is on average warmer than the spring one. Figure 3 shows the numbered precipitation stations, each number uniquely identifies
the name of each station shown in the maps below. In Figure 4, it is noted that the Region has zones of different annual
precipitation regimes. The area with the least precipitation influx is the Benevento area where it can be found the full scale
values. Almost the entire province of Naples appears to settle around an average value of about 1000 mm, in line with the
expected values, with higher values near the Lattari mountains (station number 51, 88). While in the area between the Irno
valley (station number 11, 25, 84 etc.), the Picentini Mts (station number 60), and the Sele valley (station number 26, 2) there
is a very different pluviometric regime, ranging from about 1000 to over 1400 mm on average per year. The rainiest area of
the region, at least from what emerged from this work, appears to be that of the Partenio area with values from 1500 mm
onwards, with a maximum of 2100 mm for the San Martino Valle Caudina station (number 41). In Table 4, the seasonal scale
shows that, as expected, the winter and autumn seasons strongly contribute to the accumulation of rain.  Figure 5 shows the
annual mean value of the maximum number of consecutive dry days, which, as shown by the palette on the right, range from
a minimum twenty six days to a maximum of fifty five. As it is reasonable to expect the data series of stations that recorded
the highest mean of consecutive days without rain they are found on the coastal strip, with peaks on the islands of Ischia and
Capri and in the lower Cilento area. Furthermore, it should be noted that the maximum number of consecutive dry days
decreases in the pre-Apennine and Apennine areas of the Region such as the Picentini and Partenio areas, with peaks located
in the Irpinia and upper Sannio areas. The maximum daily precipitation (Figure 6) is recorded in the mountainous areas of the
Lattari, Picentini and Partenio, and even if to a lesser extent, in that of the Matese on the border with Molise. In these areas
there is a mean maximum precipitation in one day exceeding 100 mm/days. The areas that have the lowest maximum
precipitation are those stations located in the Benevento valley with a daily maximum of 40 mm/day. In Table 4, it is noted
that in autumn the daily maximums of precipitation are on average higher than in the winter ones, for which however there is
a greater spatial variability. Moreover, the intense rainy days, obtained from the R20 indicator, vary from a minimum of two
in the summer season to a maximum of seven in the autumn. Analysing the precipitation fractions of very rainy days, it can be





seen that the 18% of summer precipitation exceeds the 95th percentile as well as in the winter season (Table 4), while the
autumn season is the one with the highest percentage of rainfall that exceeds the threshold. Looking at the rainy days (RR1) it
is noted that the wettest season is winter although the highest extreme precipitation are recorded in autumn, as evidenced by
the RX1DAY and R95PTOT indicators. The results describe a climate framework for the period 2001-2020 on the Campania
region from which it is deduced that the mean temperature it settles on an annual average between 12°C and 18°C in relation
to the geographical position of the station in question. Instead, regarding the precipitation, the wettest seasons are autumn and
winter and the areas of the region where the greatest accumulations are recorded, even daily, are the mountainous areas, in
particular the Picentini and Partenio mountains. Furthermore, the most extreme precipitations are recorded in the mountainous
areas and in the Irno valley.

|  | Yearly | ±SD | DJF | ±SD | MAM | ±SD | JJA | ±SD | SON | ±SD |
|---|---|---|---|---|---|---|---|---|---|---|
| TG °C | 14,6 | 1,6 | 7,0 | 1,7 | 13,1 | 1,6 | 22,7 | 1,5 | 15,5 | 1,6 |
| RX1DAY mm/day | 74 | 16 | 51 | 15 | 43 | 12 | 33 | 6 | 64 | 12 |
| R20 days | 19 | 6 | 6 | 3 | 4 | 2 | 2 | 0 | 7 | 2 |
| RR1 days | 97 | 9 | 33 | 3 | 27 | 3 | 11 | 2 | 26 | 2 |
| CDD days | 37 | 6 | 15 | 2 | 17 | 2 | 33 | 5 | 17 | 1 |
| R95PTOT % | 22 | 1 | 18 | 2 | 18 | 2 | 18 | 3 | 21 | 1 |
| PRCPTOT mm | 1197 | 283 | 403 | 117 | 288 | 77 | 113 | 28 | 385 | 79 |

**Table 4: Average annual and seasonal value, with evaluation of the spatial dispersion, of the climatic indicators of interest calculated on the station data series of the mean temperature and precipitation for the period 2001-2020.**

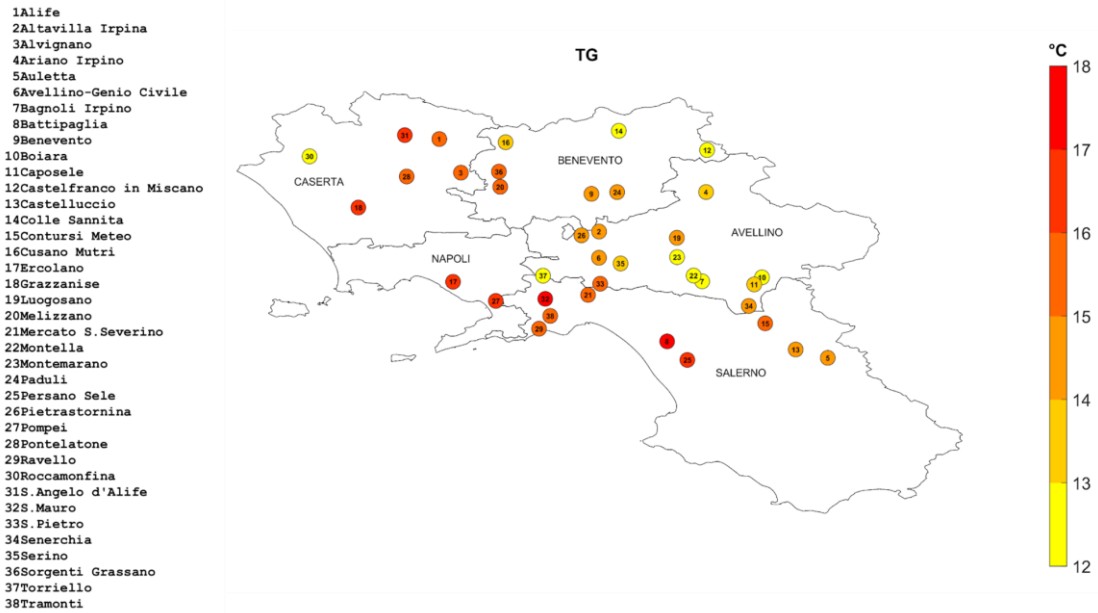

**Figure 2: Mean annual value of TG indicator over the period 2001-2020 for the Campania Region.**



| | | |
|---|---|---|
| 1Grazzanise | 31S.Felice a Cancello | 66Pietrastornina |
| 2Persano Sele | 32Pietramelara | 67Avella |
| 3Benevento | 33Liberi | 68Ponte Valentino |
| 4Montella | 34Napoli Camaldoli | 69Quattroventi |
| 5Paduli | 35Rotondi | 70S.Castrese |
| 6Auletta | 36Arienzo | 71Cologna |
| 7Castelluccio | 37Cervinara | 72Pontecagnano |
| 8Montemarano | 38S.Agata dei Goti | 73Tramonti |
| 9Senerchia | 39Ottaviano | 74Sorrento |
| 10S.Pietro | 40Visciano | 75Altavilla Irpina |
| 11S.Mauro | 41S.Martino Valle Caudina | 76Serino |
| 12Ponte Camerelle | 42Cava dei Tirreni | 77Maiori |
| 13Bellosguardo | 43Capri | 78Amalfi |
| 14Sarno | 44Massa Lubrense | 79Ravello |
| 15Cetronico | 45Corbara-S.Egidio M. | 80Ercolano |
| 16Piani di Prato | 46Pellezzano | 81Cetara |
| 17Quindici | 47Lettere | 82Agerola |
| 18Torriello | 48Torre del Greco | 83Ariano Irpino |
| 19Contursi Meteo | 49Monteforte Irpino | 84Baronissi |
| 20Alvignano | 50Solofra | 85Castiglione del Genovesi |
| 21Boiara | 51Pimonte | 86Pontelatone |
| 22Pompei | 52Mercogliano | 87Castelfranco in Miscano |
| 23S.Angelo d'Alife | 53Forino | 88Gragnano |
| 24Melizzano | 54Caserta Vecchia | 89Roccamonfina |
| 25Mercato S.Severino | 55Caiazzo | 90Sarnoo |
| 26Battipaglia | 56Napoli Capodimonte | 91Quindicii |
| 27Alife | 57Pozzuoli | 92Rofrano |
| 28Sorgenti Grassano | 58Monte Epomeo | 93Gioi Cilento |
| 29Bagnoli Irpino | 59Salerno Genio Civile | 94Montesano Terme |
| 30Caposele | 60Giffoni Valle Piana | 95S.Mauro la Bruca |
| | 61Castel Volturno | |
| | 62Avellino Genio Civile | |
| | 63Cusano Mutri | |
| | 64Colle Sannita | |
| | 65Luogosano | |

**Figure 3: List of precipitation stations in the Campania Region considered in this work.**



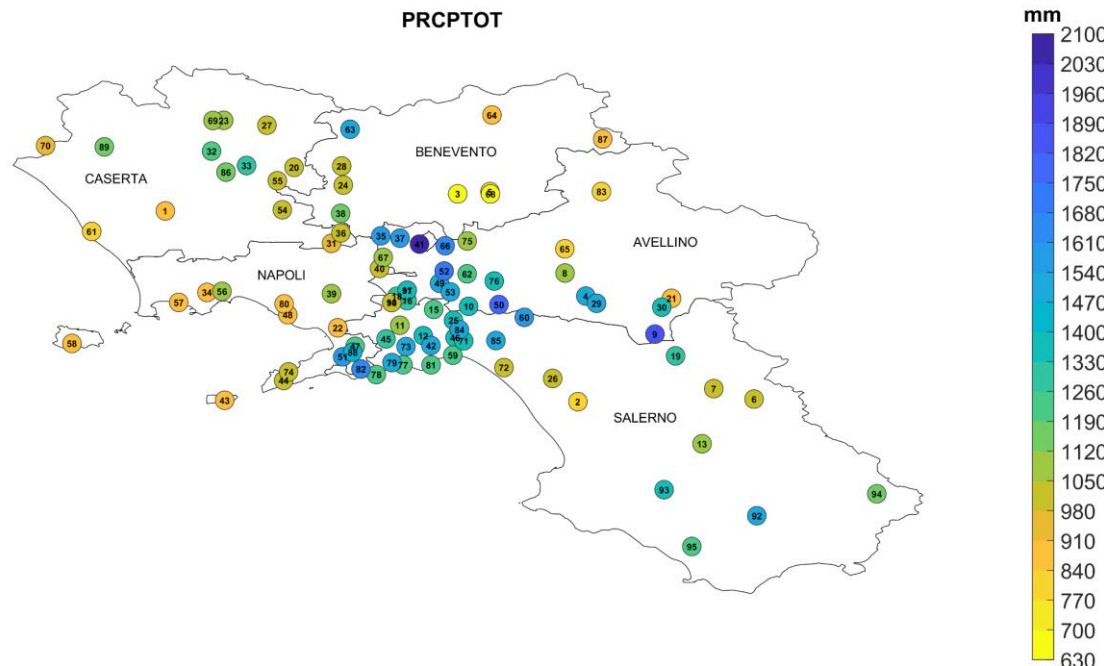

**Figure 4: Mean annual value of PRCPTOT indicator over the period 2001-2020 for the Campania Region.**

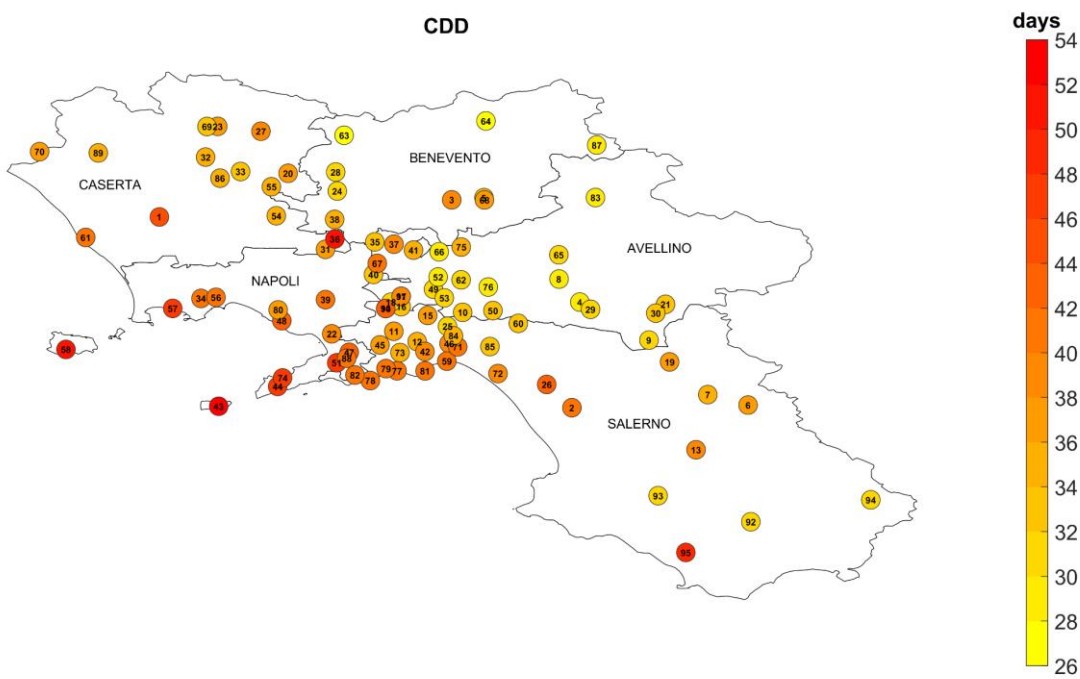

**Figure 5: Mean annual value of CDD indicator over the period 2001-2020 for the Campania Region.**





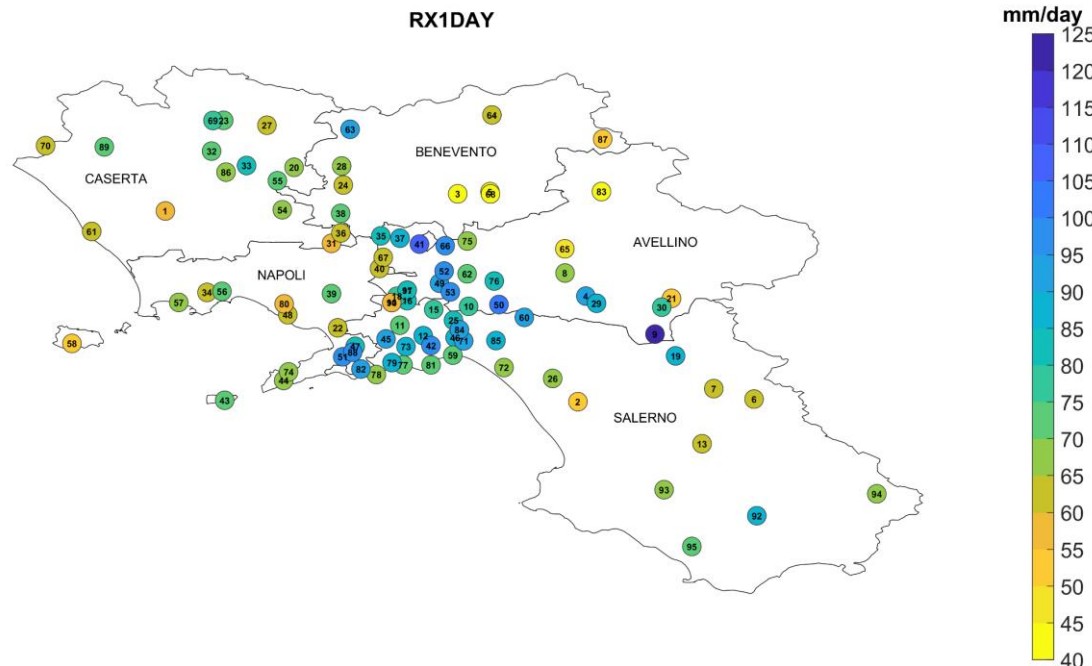

**Figure 6: Mean annual value of RX1DAY indicator over the period 2001-2020 for the Campania Region.**

## 4 Results: future climate projections

In addition to the climate framework of the Campania Region for the 2001-2020 period, necessary to characterise the current climate of the Region, this section provides the climate projections, obtained from the data simulated by the COSMO CLM model and EURO-CORDEX models, of the indicators listed in Table 3 for four different period 2021-2050, 2031-2060, 2051-2080 and 2071-2100 compared to the reference period 1981-2010, for the IPCC RCP4.5 and RCP8.5 scenarios. The climate indicators considered in this work, mainly describe the intensity and frequency of precipitation and temperature events that can be considered related to the occurrence of impacts, such as floods, cold and heat waves, fires. These analyses can then be used by subsequent sector studies aimed at evaluating the future evolution of climate change impacts on a local scale and to support climate change adaptation strategies. This section shows the results obtained on an annual scale and seasonal scale, identifying the winter season with DJF, the spring season with MAM, the summer season with JJA and the autumn season with SON. The climate anomalies of the indicators on the future thirty years of interest compared to the reference period 1981-2010 are shown below (from Table 5 to Table 8). These anomalies are obtained for each cell of the grid of the COSMO CLM model and EURO-CORDEX models, representative of the Campania Region, as the difference between the average time value of the climate indicator for the future period of interest and that relating to the reference period 1981-2010. In Table 5, the





mean temperature projections provided by COSMO CLM model show a progressive increase in the four future periods, compared to the reference period, up to an increase of 5 ° C in the long term according to the RCP8.5 scenario. On the other hand, for precipitation, an increase in the daily maximums described by the RX1DAY indicator is projected against a decrease in rainy days and therefore an increase in consecutive days without rain. It follows that a negative trend is expected for the annual cumulative precipitation, up to a reduction of 13% in the thirty years 2071-2100 for the RCP8.5 scenario. On a seasonal scale (Table 6), for the mean temperature, a significant increase is expected in all four seasons, of about 2 °C until 2060, between 2 °C and 4 °C for the period 2051-2080, up to an increase of about 6 °C for the period 2071-2100. As regards precipitation, generally, heavy rains are expected to increase in winter and autumn, while a decrease is expected in spring and summer seasons. The maximum number of consecutive days without rain increases in all seasons, up to a 50% increase in the summer season for the long-term period under the RCP8.5 scenario. The cumulative precipitation, on the other hand, shows a negative variation in spring and summer.

| | RCP4.5 | RCP8.5 | RCP4.5 | RCP8.5 | RCP4.5 | RCP8.5 | RCP4.5 | RCP8.5 |
|---|---|---|---|---|---|---|---|---|
| | Yearly | Yearly | Yearly | Yearly | Yearly | Yearly | Yearly | Yearly |
| TG °C | 1,2 | 1,5 | 1,6 | 2,1 | 2,4 | 3,6 | 2,7 | 5 |
| RX1DAY % | 7 | 10 | 9 | 13 | 14 | 13 | 13 | 21 |
| R20 % | 5 | -2 | 2 | 0 | 4 | 2 | 11 | 0 |
| RR1 % | -8 | -9 | -13 | -14 | -15 | -20 | -13 | -26 |
| CDD % | 7 | 4 | 16 | 16 | 25 | 36 | 19 | 66 |
| R95PTOT % | 3 | 3 | 4 | 5 | 6 | 7 | 6 | 10 |
| PRCPTOT % | -2 | -5 | -6 | -8 | -6 | -10 | -2 | -13 |
| | 2021-2050 vs 1981-2010 | | 2031-2060 vs 1981-2010 | | 2051-2080 vs 1981-2010 | | 2071-2100 vs 1981-2010 | |

**Table 5: Average annual anomalies of the climatic indicators of interest calculated from the data of the COSMO CLM regional model for the future periods 2021-2050, 2031-2060, 2051-2080, 2071-2100 compared to the period 1981-2010 and considering two different ones IPCC RCP4.5 and RCP8.5 scenarios.**

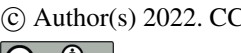



| | DJF | | MAM | | JJA | | SON | | |
|---|---|---|---|---|---|---|---|---|---|
| | RCP4.5 | RCP8.5 | RCP4.5 | RCP8.5 | RCP4.5 | RCP8.5 | RCP4.5 | RCP8.5 | |
| TG °C | 1 | 1,7 | 1,1 | 1,4 | 1,3 | 1,3 | 1,3 | 1,6 | |
| RX1DAY % | 8 | 9 | -5 | -3 | -13 | -6 | 8 | 6 | 2021-2050 vs 1981-2010 |
| R20 % | 14 | 11 | -18 | -16 | -15 | 8 | 18 | -1 | |
| RR1 % | -3 | -9 | -1 | -14 | -27 | -11 | 0 | 4 | |
| CDD % | 3 | 11 | 22 | 8 | 6 | 7 | -5 | -6 | |
| R95PTOT % | 3 | 4 | 0 | 0 | -2 | -1 | 2 | 1 | |
| PRCPTOT % | 4 | -1 | -16 | -15 | -23 | -5 | 11 | 4 | |
| TG °C | 1,6 | 2,3 | 1,3 | 1,2 | 1,9 | 2 | 1,8 | 2,3 | |
| RX1DAY % | 2 | 10 | 1 | -3 | -18 | -18 | 10 | 12 | 2031-2060 vs 1981-2010 |
| R20 % | -2 | 14 | -3 | -18 | -12 | -23 | 17 | 5 | |
| RR1 % | -13 | -15 | -15 | -17 | -35 | -29 | 1 | -2 | |
| CDD % | 8 | 24 | 21 | 3 | 11 | 15 | 1 | 2 | |
| R95PTOT % | 2 | 6 | 2 | 0 | -3 | -4 | 2 | 3 | |
| PRCPTOT % | -9 | -5 | -12 | -17 | -26 | -25 | 12 | 5 | |
| TG °C | 2,1 | 3,3 | 2,1 | 3,1 | 2,8 | 4,1 | 2,7 | 3,6 | |
| RX1DAY % | 3 | 25 | -3 | -7 | -18 | -29 | 17 | 8 | 2051-2080 vs 1981-2010 |
| R20 % | 2 | 39 | -9 | -31 | -17 | -42 | 25 | 3 | |
| RR1 % | -14 | -9 | -20 | -28 | -41 | -52 | 4 | -8 | |
| CDD % | 13 | 15 | 17 | 25 | 17 | 29 | 13 | 4 | |
| R95PTOT % | 3 | 11 | 1 | -1 | -2 | -4 | 5 | 3 | |
| PRCPTOT % | -8 | 9 | -18 | -29 | -28 | -45 | 20 | 1 | |
| TG °C | 2,4 | 4,8 | 2,5 | 4,3 | 2,9 | 5,9 | 3 | 4,9 | |
| RX1DAY % | 5 | 31 | -1 | -7 | -9 | -39 | 14 | 13 | 2071-2100 vs 1981-2010 |
| R20 % | 14 | 40 | -9 | -33 | -17 | -48 | 31 | -3 | |
| RR1 % | -8 | -10 | -19 | -36 | -32 | -67 | -1 | -20 | |
| CDD % | 4 | 12 | 15 | 41 | 16 | 50 | 15 | 17 | |
| R95PTOT % | 3 | 13 | 1 | 0 | -2 | 0 | 6 | 6 | |
| PRCPTOT % | -1 | 11 | -16 | -34 | -21 | -49 | 19 | -8 | |

377

**Table 6: Average seasonal anomalies of the climatic indicators of interest calculated from the data of the COSMO CLM regional model for the future periods 2021-2050, 2031-2060, 2051-2080, 2071-2100 compared to the period 1981-2010 and considering two different ones IPCC RCP4.5 and RCP8.5 scenarios.**

The annual anomalies calculated by EURO-CORDEX models described in Table 7 show a climate picture similar to that previously described for the COSMO CLM model. On an annual scale, Table 7 shows an increase in the phenomena of intense precipitation, in fact a general increase in the RX1DAY, R20 and R95PTOT indicators is expected. The projections also identify an increase in consecutive dry days, and a general constant decrease in the cumulative annual precipitation. As regards the mean temperature, a very significant growth trend is identified, reaching even 4 °C in the thirty years 2071-2100 for the RCP8.5 scenario.

To avoid compensation phenomena during the year, the seasonal scale is also analysed in Table 8. There is an always constant increase in temperatures over time, but less sudden, especially more marked in the spring and summer seasons. For precipitation, on the other hand, an increase in intense precipitation is expected in winter, while a decrease in rainy days, as has just been described for both RCP45 and RCP85, the latter with more pronounced values; consecutive days without rain





will increase in the spring and summer season and the projection of the RCP8.5 scenario to the thirty-year period 2071-2100
assumes a 22% decrease on the cumulative rainfall in summer. For the autumn season also in this case the projections show a
decrease in rainy days but an increase in intense precipitation. Furthermore, Tables 7 and 8 show greater agreement between
the EURO-CORDEX models in the short and medium term periods for both scenarios.

| | RCP4.5 | | RCP8.5 | | RCP4.5 | | RCP8.5 | | RCP4.5 | | RCP8.5 | | RCP4.5 | | RCP8.5 | |
|---|---|---|---|---|---|---|---|---|---|---|---|---|---|---|---|---|
| | Yearly | ±SD | Yearly | ±SD | Yearly | ±SD | Yearly | ±SD | Yearly | ±SD | Yearly | ±SD | Yearly | ±SD | Yearly | ±SD |
| TG °C | 1 | 0 | 1 | 0 | 1,3 | 0 | 1,7 | 0 | 1,7 | 0,4 | 2,7 | 0,4 | 2 | 0 | 4 | 1 |
| RX1DAY % | 3 | 6 | 6 | 6 | 5 | 6 | 7 | 6 | 9 | 5 | 11 | 7 | 10 | 6 | 15 | 9 |
| R20 % | 3 | 9 | 2 | 8 | 2 | 8 | 3 | 6 | 3 | 5 | 6 | 7 | 6 | 7 | 1 | 11 |
| RR1 % | -3 | 3 | -4 | 4 | -4 | 3 | -6 | 4 | -5 | 4 | -9 | 5 | -6 | 5 | -15 | 7 |
| CDD % | 6 | 9 | 5 | 8 | 9 | 9 | 7 | 8 | 9 | 8 | 15 | 10 | 9 | 10 | 25 | 19 |
| R95PTOT % | 2 | 2 | 2 | 2 | 2 | 2 | 3 | 2 | 3 | 2 | 5 | 3 | 4 | 2 | 6 | 3 |
| PRCPTOT % | -1 | 5 | -1 | 6 | -2 | 4 | -2 | 6 | -2 | 4 | -3 | 7 | 0 | 6 | -8 | 12 |
| | 2021-2050 vs 1981-2010 | | | | 2031-2060 vs 1981-2010 | | | | 2051-2080 vs 1981-2010 | | | | 2071-2100 vs 1981-2010 | | | |


**Table 7: Average annual anomalies, with evaluation of the uncertainty, of the climatic indicators of interest calculated from the data**
**of the EURO-CORDEX models for the future periods 2021-2050, 2031-2060, 2051-2080, 2071-2100 compared to the period 1981-**
**2010 and considering two different ones IPCC RCP4.5 and RCP8.5 scenarios.**

| | DJF | | | | MAM | | | | JJA | | | | SON | | | | |
|---|---|---|---|---|---|---|---|---|---|---|---|---|---|---|---|---|---|
| | RCP4.5 | ±SD | RCP8.5 | ±SD | RCP4.5 | ±SD | RCP8.5 | ±SD | RCP4.5 | ±SD | RCP8.5 | ±SD | RCP4.5 | ±SD | RCP8.5 | ±SD | |
| TG °C | 0,9 | 0 | 1 | 0 | 0,8 | 0 | 1 | 0 | 1,4 | 0 | 1,5 | 0 | 0,9 | 0 | 1,3 | 0 | |
| RX1DAY % | 4 | 6 | 3 | 9 | 1 | 6 | -1 | 7 | -4 | 12 | -1 | 14 | 6 | 8 | 8 | 5 | 2021-2050 vs 1981-2010 |
| R20 % | 7 | 12 | 5 | 17 | 0 | 16 | -4 | 15 | -7 | 27 | -2 | 23 | 7 | 13 | 7 | 6 | |
| RR1 % | -1 | 6 | -3 | 8 | -3 | 4 | -5 | 8 | -10 | 11 | -10 | 9 | 0 | 7 | -1 | 5 | |
| CDD % | 1 | 10 | 4 | 15 | 4 | 6 | 6 | 11 | 7 | 11 | 4 | 8 | 0 | 12 | 1 | 9 | |
| R95PTOT % | 1 | 2 | 1 | 3 | 0 | 2 | 0 | 2 | 0 | 2 | 0 | 3 | 2 | 3 | 3 | 2 | |
| PRCPTOT % | 1 | 8 | -1 | 11 | -3 | 8 | -5 | 11 | -11 | 14 | -9 | 15 | 4 | 11 | 3 | 4 | |
| TG °C | 1,2 | 0 | 1,5 | 0 | 1 | 0 | 1,4 | 0 | 1,8 | 0 | 2,1 | 0 | 1,2 | 0 | 1,8 | 0 | |
| RX1DAY % | 4 | 6 | 5 | 7 | 0 | 5 | -1 | 6 | -5 | 15 | -3 | 15 | 8 | 9 | 9 | 7 | 2031-2060 vs 1981-2010 |
| R20 % | 7 | 10 | 7 | 13 | -4 | 15 | -5 | 12 | -8 | 33 | -3 | 27 | 9 | 14 | 9 | 8 | |
| RR1 % | -2 | 4 | -5 | 7 | -6 | 5 | -8 | 7 | -14 | 11 | -12 | 12 | 0 | 8 | -3 | 5 | |
| CDD % | 1 | 8 | 7 | 13 | 4 | 7 | 9 | 10 | 10 | 10 | 6 | 8 | -1 | 11 | 5 | 9 | |
| R95PTOT % | 1 | 2 | 2 | 3 | 0 | 2 | 0 | 2 | 0 | 3 | 0 | 4 | 3 | 3 | 4 | 3 | |
| PRCPTOT % | 0 | 36 | -2 | 8 | -6 | 8 | -7 | 9 | -14 | 15 | -11 | 19 | 5 | 11 | 4 | 6 | |
| TG °C | 1,5 | 0,4 | 2,3 | 0,4 | 1,4 | 0,4 | 2,3 | 0,4 | 2,1 | 0,4 | 3,3 | 0,7 | 1,7 | 0,5 | 2,7 | 0,6 | |
| RX1DAY % | 4 | 6 | 8 | 7 | 0 | 8 | 1 | 7 | -2 | 18 | -1 | 16 | 13 | 8 | 14 | 7 | 2051-2080 vs 1981-2010 |
| R20 % | 4 | 9 | 12 | 8 | -3 | 16 | -4 | 13 | 0 | 37 | -3 | 27 | 11 | 11 | 13 | 11 | |
| RR1 % | -4 | 5 | -6 | 6 | -7 | 7 | -13 | 6 | -12 | 10 | -18 | 16 | -1 | 8 | -6 | 8 | |
| CDD % | 5 | 9 | 8 | 13 | 4 | 6 | 12 | 11 | 9 | 8 | 13 | 12 | 3 | 13 | 7 | 9 | |
| R95PTOT % | 2 | 2 | 4 | 3 | 0 | 2 | 1 | 2 | 0 | 4 | 1 | 4 | 4 | 3 | 6 | 2 | |
| PRCPTOT % | -2 | 6 | -1 | 5 | -6 | 10 | -12 | 7 | -10 | 17 | -14 | 24 | 6 | 10 | 4 | 9 | |
| TG °C | 1,8 | 0 | 3,3 | 0 | 1,7 | 0 | 3,3 | 1 | 2,4 | 1 | 4,6 | 1 | 2 | 0 | 3,9 | 1 | |
| RX1DAY % | 7 | 5 | 8 | 8 | 0 | 7 | 1 | 7 | 0 | 17 | -6 | 25 | 14 | 7 | 18 | 7 | 2071-2100 vs 1981-2010 |
| R20 % | 10 | 7 | 7 | 13 | -2 | 13 | -8 | 14 | 2 | 31 | -15 | 38 | 12 | 9 | 7 | 8 | |
| RR1 % | -4 | 5 | -11 | 7 | -9 | 7 | -19 | 8 | -10 | 13 | -28 | 20 | -2 | 6 | -11 | 8 | |
| CDD % | 4 | 10 | 12 | 15 | 11 | 8 | 22 | 10 | 8 | 10 | 23 | 19 | 3 | 11 | 9 | 14 | |
| R95PTOT % | 3 | 2 | 4 | 3 | 1 | 2 | 2 | 2 | 1 | 4 | 1 | 6 | 5 | 3 | 7 | 2 | |
| PRCPTOT % | 1 | 6 | -5 | 8 | -7 | 8 | -17 | 9 | -7 | 20 | -22 | 35 | 6 | 6 | -1 | 9 | |






**Table 8: Average seasonal anomalies, with evaluation of the uncertainty, of the climatic indicators of interest calculated from the**
**data of the EURO-CORDEX models for the future periods 2021-2050, 2031-2060, 2051-2080, 2071-2100 compared to the period**
**1981-2010 and considering two different ones IPCC RCP4.5 and RCP8.5 scenarios.**

The climate framework shown by the climate projections highlights a future decrease in the cumulative total precipitation
(PRCPTOT) and in the rainy days (RR1), especially in the summer season. On the other hand, an increase in consecutive days
without precipitation, days of heavy rain and the fraction of rain due to precipitation above the $95^{th}$ percentile in the winter
and autumn seasons is expected. It is also projected an increase in the daily maximums of precipitation in winter and autumn,
while a general decrease in the spring and summer seasons. The increase in consecutive days without precipitation and the
decrease in total precipitation, together with the increase in extreme values, suggest that short-term but high intensity
precipitation events are expected in the future. Regarding mean temperature, both scenarios and models agree on a future
increase over the whole region.
**5 Conclusions**
The climate framework shown by the indicators calculated on the data of the meteorological station networks of the Campania
region for the period 2001-2020, showed that the annual mean temperature map highlights the great dependence on the
orography of the territory as lower mean temperature values are recorded for stations located in the innermost areas of the
region. Again, the greatest accumulations of precipitation are concentrated in the main mountain areas, and in particular the
stations located in the mountainous area of Partenio are those that recorded the highest values. Furthermore, these same areas
are the ones most affected by heavy rain. Autumn and winter, as you might expect, are the seasons for which the largest
accumulations of precipitation and the greatest number of intense events occur. The expected climate situation for the
Campania region compared to the reference period 1981-2010 projected an increase in the mean temperature; in terms of
precipitation it expected an increase in the maximum number of consecutive dry days. Furthermore, a rather general decrease
in total precipitation is awaited, on the other hand, an increase in intense events is projected, especially in the autumn and
winter season. From a comparison between the results on the observed climate and on the expected climate in the future period
under review, it can be deduced that as total cumulative precipitation decreases, heavy rain events increase, especially in the
autumn and winter season. This implies that total rainfall accumulations over the region will be caused by shorter time-scale
events causing rainwater disposal problems on rainy days, and drought problems during periods when rains run out.





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
