# Peer review of "Past and future climate analysis at regional scale: the case study of the Campania"

_EGUsphere, 2022_

## Author Comment (AC1)

Dear Reviewer 1,

Many thanks for taking the time to read our manuscript and provide helpful feedback. It is very much appreciated.

We have copied your comments below, and responded to them after each comment

Best regards
The Authors.

Major revision

A) My first major point is the structure of the paper. A general paper structure is there but I think that some parts are not filled with the right content or important parts are missing.

The Introduction is missing of some content in my point of view. Introductions include a literature review or summary of the current research on that field, and this is not included. I would suggest including more paragraphs that answer questions like: why are you doing it, what has been done before, how does this study fit to published studies, what can/will this study add? Same for the conclusion. In my opinion the conclusion also includes some discussion part which is missing. Do these results fit to other research, do they show same or other results than studies for other regions or the same region? Please, add some paragraphs that put your study in connection to the current available literature. I did find a few examples which could fit from the title (didn't read them totally)

Also, there is no Data chapter. I believe starting with a data section before introducing the method would help the reader to understand with what data you are working and maybe what you try to do with it. I believe most of the data description is in the manuscript already but very scattered and not in a good following order. E.g., most of chapter 2.3 is more data than Method. You could also easily use Fig. 1 in the data chapter and describe the region and stations. Also Fig. 1 is not named in the manuscript. Please, check that all figures, tables and references you have been named in the text at least once. Also, when Data is described before Method, all the climatic indicators of temperature and rain could be explained there, and this knowledge would make the method part less theoretical.

Many thanks for the advice, we tried to add, as you gently suggested, a chapter about the Data we use for the paper, furthermore we think that was a good idea add a paragraph about the geographical and climatological description of the Region. We modified the introduction following your suggestion, expanding it by adding references to previous works with similar themes to the one dealt with here.

B) My second major point is the Method section. This part is very theoretical and described in a complex way. Especially section 2.1.3 and 2.2.3 were not really clear in my opinion while reading it. I would suggest a way shorter description of each test with the respective references and maybe add these additional and more complex descriptions to the supplementary material. These tests are nothing new, developed by you, if I understand correctly, so if people want to use it, they can look them up in the original source. I believe there is not much need for the equations, maybe also something for the supplementary material. In connection to my point A), I was a bit confused what these tests are for, because I didn't really have an overview of the study. I believe have a bit more information in the Introduction and a Data section before could help.

We fully agree with this second point, the section of the methods has been summarized to a simple description of all the tests carried out with references to the original references.

C) Another part is the results chapter. I see the interest in this study, and I believe it can be of relevance, but I think there is more information needed, more background information and more structure to the results. What is the new aspect of the results chapter if most of the results are expected? You connect it 2 or 3 times to floods and droughts, but a bit more discussion in this direction would make it a better fit into NHESS.

I would classify Figure 3 as a table. My recommendation would be to give the stations the same number over temperature and precipitation and have this table in the mentioned Data chapter with all stations, their numbers and maybe even altitudes.

For the climate change anomaly results, could you add details about the significance of the anomalies. I believe a simple t-test would be sufficient and could be added into the tables with a symbol like "*".

A recommendation for all tables with data in it: maybe colour the cells according to their value, so reader can easily see negatives in blue, and positives in red (as example). Maybe even change the shade like the minimum is the strongest blue and the maximum the strongest red per climatic indicator

Thank you for the suggestion, to assess the statistically significant trend of the climate anomalies of the indicators, the Mann-Kendall test was used with a 95% confidence level. In section "5.2 Future climate projections", in all the Tables, the anomaly value with statistically significant trend is identified by an asterisk.

D) Please, check the reference for consistence. I found in the reference list some references which are not used in the manuscript. One author has 2 references of the same year, so they need a & b, others had an a on the year but only existed ones in the list, so the a is not needed. Also please stay with a continues references style, best the one the journal recommends.

Thank you, we have reviewed the entire paragraph of the references and corrected what you wrote

E) My English is not perfect either, but I have the impression this needs to be reviewed for correct language.

We apologize for the difficulties encountered in reading the document, we tried to revise the article to correct the language

---

## Author Comment (AC2)

Dear Reviewer 2,

Many thanks for taking the time to read our manuscript and provide helpful feedback. It is very much appreciated.

We have copied your comments below, and responded to them after each comment

Best regards
The Authors.

General Issue

ETCCDI provides a list of 27 indicators, it is not clear why authors have used only the limited subset shown in Table 3. In particular, only one indicator for temperature is not sufficient.

This article presents a selection of ETCCDI indicators that have been carried out as part of a specific research activity. In particular, suitable climatic indicators have been identified, starting from literature studies, to provide information on the danger due to climate change in the following areas: state and availability of the water resource, geological, hydrogeological and hydraulic instability.

A wide part of the manuscript (about seven pages) is devoted to the description of the methods used for completeness and homogenization tests, but these techniques are well established in literature, they can be easily found in books, but also in multimedia channel, e.g:

- https://www.youtube.com/watch?v=KQsshJ04WxM
- https://rdrr.io/cran/trend/man/br.test.html

so, there is no need to describe them in details. They can be only mentioned with their pro and cons, while the full description could be replaced by proper references.

Thanks for the advice, we appreciate that. We already provide to summarize this part of the manuscript trying to avoid prosaic sentences; we simply left a briefly description of the methods and the original references are reported

The analysis of past climate, although formally correct, looks just as a description of numbers, but no scientific interpretation is provided, apart from some prosaic and obvious sentences. Some unusual behaviors have been observed, but no physical interpretation or justification is given.

We briefly added a comparison between the result obtained in this manuscript and the results of other papers already published.

The analysis of future climate projections is pretty modest. The authors do not explain the reason why these two models (cosmo-clm and eurocordex ensemble mean) and these two RCP scenarios have been selected. Also, before performing future analysis, the model must be validated against observational data, in order to assess the capabilities of the models in reproducing the climate features of the area under study. Biases

affecting simulations must be quantified. Finally, a critical analysis of the projections is completely missing, as well as comparison with other available projections (even at lower resolutions) to check their consistency.

The choice of the two models used (cosmo-clm and eurocordex ensemble mean) depends on the fact that both are validated in the literature (see subsections 3.2 and 3.3). Furthermore, the EURO-CORDEX models have been used because thanks to the use of a set of climate models, it is also possible to associate the expected climate changes with an analysis of uncertainty, a very important element for climate adaptation and risk analysis.